# A novel enhancer near the *Pitx1* gene influences development and evolution of pelvic appendages in vertebrates

Abbey C Thompson[1,2], Terence D Capellini[1†], Catherine A Guenther[1,3], Yingguang Frank Chan[1‡], Carlos R Infante[4§], Douglas B Menke[4], David M Kingsley[1,3]*

[1]Department of Developmental Biology, Stanford University School of Medicine, California, United States; [2]Department of Genetics, Stanford University School of Medicine, California, United States; [3]Howard Hughes Medical Institute, Stanford University, California, United States; [4]Department of Genetics, University of Georgia, Georgia, United States

*For correspondence:
kingsley@stanford.edu

Present address: [†]Human Evolutionary Biology, Harvard University, Cambridge, United States; [‡]Friedrich Miescher Laboratory, Max Planck Society, Tübingen, Germany; [§]Department of Integrative Biology, University of Colorado Denver, Denver, United States

Competing interests: The authors declare that no competing interests exist.

**Abstract** Vertebrate pelvic reduction is a classic example of repeated evolution. Recurrent loss of pelvic appendages in sticklebacks has previously been linked to natural mutations in a pelvic enhancer that maps upstream of *Pitx1*. The sequence of this upstream *PelA* enhancer is not conserved to mammals, so we have surveyed a large region surrounding the mouse *Pitx1* gene for other possible hind limb control sequences. Here we identify a new pelvic enhancer, *PelB*, that maps downstream rather than upstream of *Pitx1*. *PelB* drives expression in the posterior portion of the developing hind limb, and deleting the sequence from mice alters the size of several hind limb structures. *PelB* sequences are broadly conserved from fish to mammals. A wild stickleback population lacking the pelvis has an insertion/deletion mutation that disrupts the structure and function of *PelB*, suggesting that changes in this ancient enhancer contribute to evolutionary modification of pelvic appendages in nature.
DOI: https://doi.org/10.7554/eLife.38555.001

## Introduction

Vertebrate limbs and fins show remarkable morphological diversity (*Flower, 1870*; *Hinchliffe and Johnson, 1980*). Extensive limb modifications are seen in animals adapted to running, jumping, swimming, flying, digging, and hunting or evading predators. Of particular interest are species that have undergone dramatic modifications to one set of paired appendages while leaving the other set largely unmodified. For example, bats, birds, and flying fish have greatly elongated pectoral appendages compared to pelvic appendages. Conversely, whales, manatees, and several fish groups have reduced or lost their pelvic appendages, while retaining robust pectoral appendages. Despite long-standing interest in the processes that lead to species-specific limb differences (*Owen, 1849*), the detailed genetic changes that underlie vertebrate limb modifications are still largely unknown.

Numerous signaling and transcription factor pathways have been identified that play a key role in limb development (reviewed in *Zuniga, 2015*; *Petit et al., 2017*). Though most of these factors are expressed in both the fore- and hind limbs, some are limb-specific. *Tbx5* is expressed in the developing forelimb (*Gibson-Brown et al., 1996*), and complete loss of this gene prevents proper forelimb outgrowth (*Ahn et al., 2002*; *Garrity et al., 2002*). By contrast, *Tbx4* is expressed specifically in the developing hind limb (*Chapman et al., 1996*), and loss of one or both copies results in significant hind limb defects (*Naiche and Papaioannou, 2003*; *Bongers et al., 2004*; *Naiche and Papaioannou, 2007*).

Another hind limb-specific gene is *Pitx1*, which encodes a homeodomain transcription factor that acts upstream of *Tbx4* (*Logan and Tabin, 1999*). The hind limb-specific expression pattern of *Pitx1* is conserved in many different vertebrates, including fish, birds, and mammals (*Lanctôt et al., 1997*; *Logan et al., 1998*; *Shapiro et al., 2006*). *Pitx1* is clearly required for normal hind limb development in mammals, as *Pitx1* knockout mice show reduced hind limb size, complete loss of the ilium and patella, and altered shape of the remaining hind limb bones (*Lanctôt et al., 1999*; *Szeto et al., 1999*). Mutations in the *Pitx1* gene or its surrounding regulatory regions are also associated with avian foot feathering (*Domyan et al., 2016*) and several human limb abnormalities, including polydactyly (*Klopocki et al., 2012*), Liebenberg syndrome (*Spielmann et al., 2012*), and familial clubfoot (*Gurnett et al., 2008*). In addition to its role in hind limb development, the *Pitx1* gene is also expressed and required in several other tissues. Mice with a complete knockout of *Pitx1* die soon after birth with pituitary gland abnormalities, shortened jaws, and cleft palate (*Lanctôt et al., 1999*; *Szeto et al., 1999*).

Although complete loss of *Pitx1* is clearly deleterious, previous studies suggest that regulatory changes in *Pitx1* have also contributed to adaptive evolution of new skeletal traits in wild species. Genetic crosses between wild stickleback fish have shown that *Pitx1* is a major effect locus controlling pelvic reduction that has evolved repeatedly in many freshwater populations (*Cresko et al., 2004*; *Shapiro et al., 2004*; *Coyle et al., 2007*). High-resolution genetic mapping and enhancer studies identified a non-coding regulatory sequence located upstream of *Pitx1*, called *Pel*, that drives expression specifically in the developing pelvic hind fin (*Chan et al., 2010*). The *Pel* enhancer has been independently deleted in many pelvic-reduced freshwater populations, accompanied by molecular signatures of positive selection centered around the *Pel* deletions (*Chan et al., 2010*). Reintroduction of *Pel*-driven *Pitx1* can restore pelvic development in pelvic-reduced sticklebacks, providing strong evidence that regulatory changes in *Pitx1* underlie the repeated loss of pelvic hind fins in wild sticklebacks (*Chan et al., 2010*).

Although the hind limb-specific expression pattern of *Pitx1* is conserved across vertebrates, the primary sequence of the stickleback *Pel* enhancer is not. *Pel* enhancer orthologs can be identified in other fish, but not in most other vertebrates, including mammals (*Chan et al., 2010*). Conversely, genetic studies in birds, mice, and humans suggest multiple upstream regions may be involved in *Pitx1* limb expression, but it has been difficult to identify any individual *Pitx1* enhancers that drive hind limb-specific expression (*Pennacchio et al., 2006*; *Spielmann et al., 2012*; *Domyan et al., 2016*; *Kragesteen et al., 2018*; *Sarro et al., 2018*). To identify possible mammalian hind limb enhancers in *Pitx1*, we have now surveyed a large region surrounding the mouse *Pitx1* locus for sequences that drive limb expression in transgenic mice. Here we identify a novel pelvic enhancer located downstream rather than upstream of *Pitx1*, which shows conservation of both sequence and function from mammals to fish. Genetic studies suggest this sequence influences development of a subset of normal hind limb features in mice, and has also contributed to evolutionary pelvic reduction in natural populations.

## Results

### A BAC scan across the *Pitx1* locus

To identify regulatory sequences that drive limb expression in developing mouse embryos, we surveyed an 850 kb region of the *Pitx1* locus using overlapping mouse BAC clones that cover the entire *Pitx1* coding region as well as flanking regions (*Figure 1*). Note that the scan includes 607 kb of sequence upstream of the *Pitx1* coding region. The known *Pel* enhancer maps upstream of the stickleback *Pitx1* gene (*Chan et al., 2010*), and although a mammalian *Pel* ortholog cannot be identified by sequence alignment, it is possible that a functionally conserved *Pel* enhancer also resides in the same upstream region in mammals. Two large deletions associated with human forelimb abnormalities also map upstream of the *Pitx1* gene (*Figure 1*). Both deletions remove the flanking *H2afy* gene and bring a far upstream enhancer called *hs1473* closer to the human *Pitx1* gene (*Spielmann et al., 2012*). The *hs1473* enhancer can drive gene expression in both forelimbs and hind limbs (*Pennacchio et al., 2006*), and has hence been named the *pan-limb enhancer* or *Pen* (*Kragesteen et al., 2018*). Ectopic expression of *Pitx1* in forelimbs likely causes the arm-to-leg-like morphological abnormalities characteristic of human Liebenberg syndrome (*Spielmann et al., 2012*;

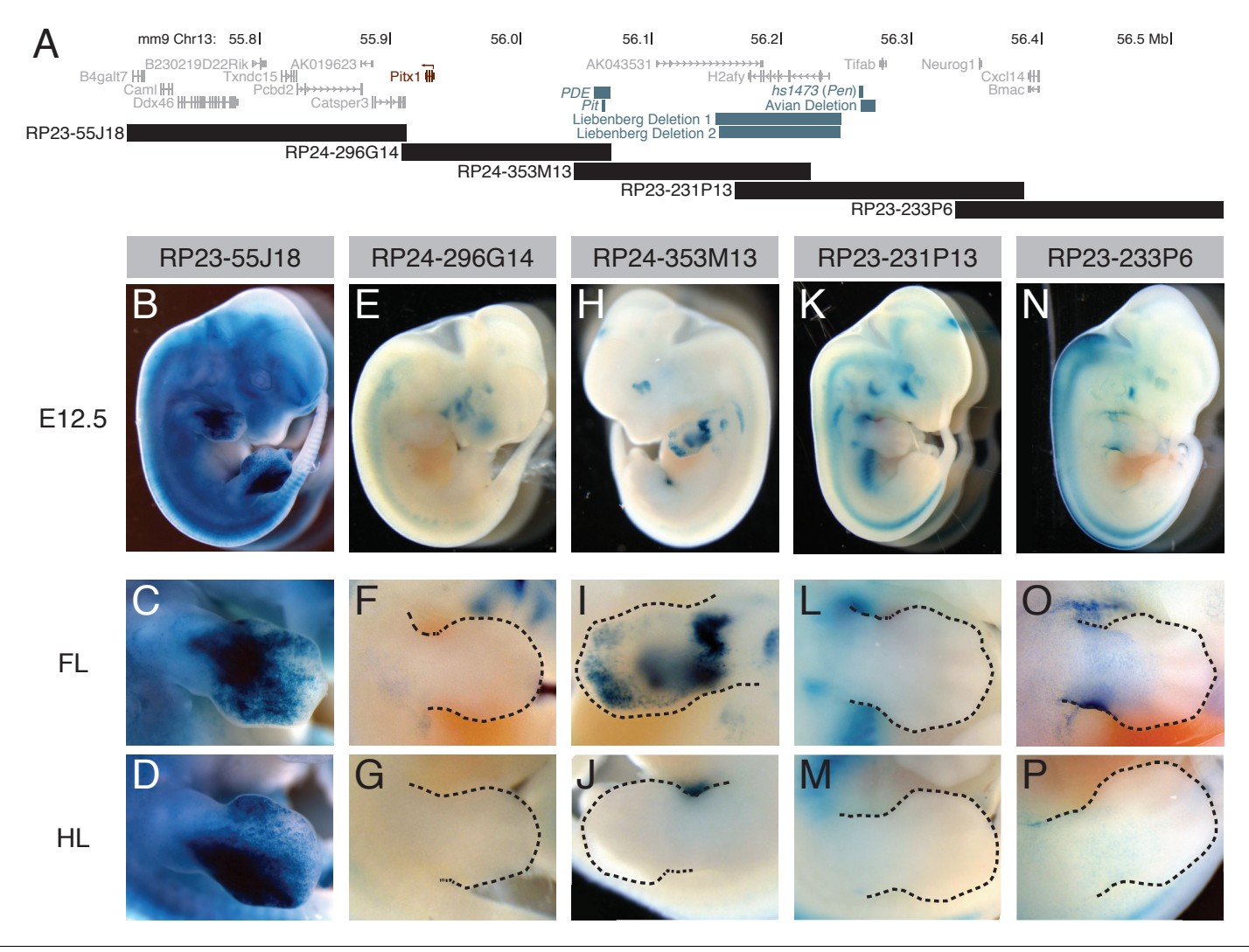

**Figure 1.** A BAC scan across the *Pitx1* locus. (**A**) Schematic of a genomic region surrounding *Pitx1* (red). Note that the transcriptional orientation of the *Pitx1* gene is from right to left in this view (arrow). Black bars show BAC locations. Gray bars denote orthologous positions of other known or suspected *cis* regulatory regions, including: sequences removed by the avian deletion associated with feathered feet in pigeons (*Domyan et al., 2016*), the *PDE* element (*Sarro et al., 2018*), the *Pit* enhancer (*Kragesteen et al., 2018*), human deletions associated with Liebenberg syndrome (*Spielmann et al., 2012*), and the *hs1473* limb enhancer (*Pennacchio et al., 2006*; *Spielmann et al., 2012*), now also referred to as the *pan-limb enhancer*, or *Pen* (*Kragesteen et al., 2018*). (**B-P**) Whole-mount lacZ staining of transgenic embryos carrying RP23-55J18 (**B-D**), RP24-296G14 (**E-G**), RP24-353M13 (**H-J**, only left side of embryo showed limb staining), RP23-231P13 (**K-M**), or RP23-233P6 (**N-P**), with close ups of the forelimb ('FL') and hind limb ('HL') below. Dotted lines denote boundaries of limbs.

DOI: https://doi.org/10.7554/eLife.38555.002

*Kragesteen et al., 2018*). A deletion associated with lowered *Pitx1* expression in hind limbs and feathered feet in birds also maps upstream of the *H2afy* gene (*Domyan et al., 2016*).

In our initial screen, two different BAC clones drove lacZ expression in the pituitary gland, a well-known site of endogenous *Pitx1* expression (BACs RP24-296G14 and RP24-353M13, see *Supplementary file 1A*). These two BACs overlap in a 29 kb region, suggesting this shared region may contain a pituitary enhancer. Interestingly, this shared region also contains a sequence called the *Pitx1 Distal Enhancer* (*PDE*, *Figure 1*), a region that shows significant chromatin interactions in limb tissues with the promoter of *Pitx1* (*Sarro et al., 2018*). This region has been knocked out in mice, leading to modest reductions in *Pitx1* expression in limbs and mandibles, but no detectable limb or jaw skeletal phenotypes (*Sarro et al., 2018*). Possible effects on pituitary development and

expression were not reported in *PDE* knockout mice. However, a 2.5 kb subregion of the *PDE* region can drive consistent expression in the developing pituitary, and this region has now been named the *Pit* enhancer (*Kragesteen et al., 2018*).

Three different BAC clones drove lacZ expression in the developing forelimb (*Figure 1C,I,O*). However, only one of these clones, BAC RP23-55J18, also drove prominent expression in the hind limb. With this clone, lacZ expression was observed at the proximal end of the forelimb autopod (*Figure 1C*), as well as the posterior side of the hind limb autopod (*Figure 1D*). Although the expression driven by RP23-55J18 was not specific to the hind limb, it is possible that multiple limb enhancers exist in the *Pitx1* region, and that the overall pattern represents the combined activity of separate forelimb and hind limb enhancers.

## Isolation of a *Pitx1* pelvic hind limb enhancer

Previous studies suggest that evolutionarily conserved sequences and tissue-specific chromatin marks can often be used to identify tissue-specific enhancers (*Fortini and Rubin, 1990*; *Mortlock et al., 2003*; *Woolfe et al., 2005*; *Pennacchio et al., 2006*). To further test for possible hind limb enhancer regions within the interval covered by BAC RP23-55J18, we looked for conserved regions that also showed increased chromatin accessibility or H3K27ac marks in hind limb tissues relative to forelimb (*Figure 2* and *Figure 2—figure supplement 1*). We subcloned several of these regions upstream of a lacZ reporter and tested whether these sequences were capable of driving reporter gene activity at consistent locations in transgenic mice.

A 9466 bp region downstream of *Pitx1*, and located within a large intron of the *Pcbd2* gene, contains multiple conserved sequences with increased DNaseI accessibility during hind limb development (*Figure 2A*). Chromosome conformation capture experiments (*Andrey et al., 2017*) show that this region directly interacts with the *Pitx1* promoter in hind limbs but not forelimbs of developing mouse embryos, consistent with the region serving as a possible *Pitx1* enhancer (*Supplementary file 1B*). The same region is enriched for H3K27ac signal in developing hind limbs versus forelimbs of both mouse and lizard embryos (*Infante et al., 2015*), suggesting the region may contain evolutionarily conserved hind limb enhancer activity (*Figure 2—figure supplement 1*). A lacZ expression construct containing this region drove strong and reproducible lacZ expression in the hind limbs but not forelimbs of transient transgenic E12.5 embryos (*Figure 2B*). Strongest expression was observed in the posterior half of the developing autopod, with weaker expression also seen at the proximal junction of hind limb and body wall. We designated the full 9466 bp fragment *pelvic limb enhancer B (PelB)*. A 2173 bp subregion (*PelBcon1*), containing a hind limb-enriched open chromatin domain and sequence conservation through amphibia, exhibited the same proximal expression as the full-length fragment (*Figure 2C*). A separate, non-overlapping 3280 bp subregion (*PelBcon2*), with a strong hind limb-enriched DNase peak and sequence conservation through teleosts, exhibited the same autopod pattern as the full-length fragment (*Figure 2D*).

## Deletion of *PelB* enhancer in mouse

To test whether *PelB* is required for normal hind limb development, we used CRISPR/Cas9 targeting to delete a 9425 bp region from the endogenous mouse locus, encompassing virtually the entire *PelB* region tested in the lacZ assay (see Materials and methods). The deletion allele was transmitted through the germline, and subsequent crosses showed that *PelB* heterozygotes and homozygotes were born in expected Mendelian ratios, and showed normal viability and fertility. Quantitative RT-PCR experiments of developing wild type and *PelB* homozygous mutant embryos showed that loss of the *PelB* enhancer reduced *Pitx1* levels to approximately 85% of control levels in E12.5 hind limbs (p<0.05, *Supplementary file 1C*). In contrast, no significant change was seen in *Pcbd2* expression (*Supplementary file 1C*). These results suggest that *PelB* acts as an enhancer of the *Pitx1* gene during normal development, but that additional control regions also must contribute to overall levels of *Pitx1* hind limb expression.

To identify possible effects of *PelB* enhancer loss on hind limb development, we analyzed bones of adult *Pitx1^PelB-/PelB-* mice and control mice. One *Pitx1^PelB-/PelB-* mouse exhibited right hind limb preaxial polydactyly, an alteration known to occur at low background rates in the C57BL/6J mouse background (*Dagg, 1966*). This mouse was excluded from further analysis. No other major changes in the number, or presence or absence of hind limb, forelimb, or jaw bones were apparent in

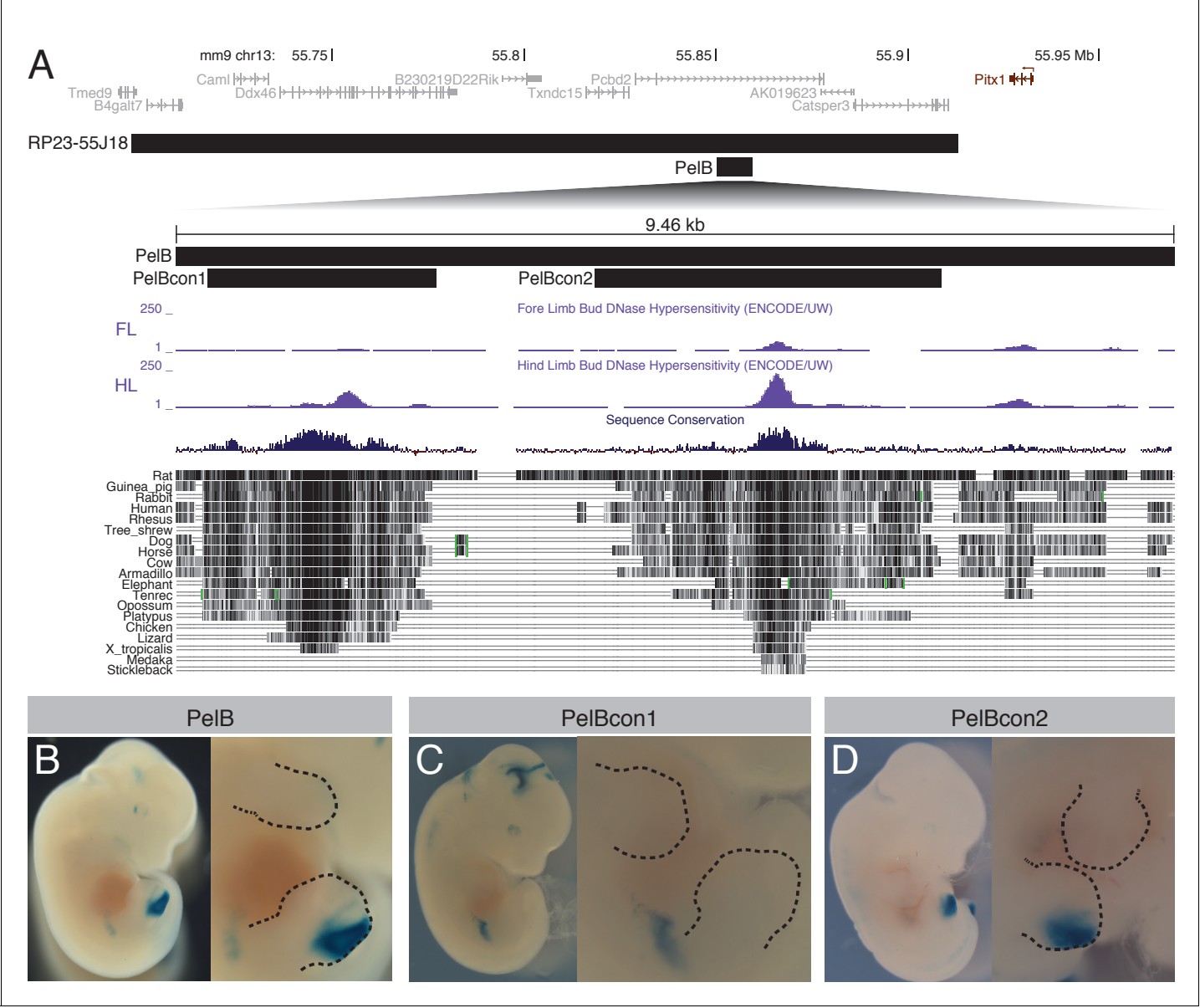

**Figure 2.** A novel pelvic enhancer downstream of *Pitx1*. (**A**) Schematic of genomic region encompassing *Pitx1* (black), with flanking genes (gray). Black bars mark the location of the BAC RP23-55J18 and smaller enhancer constructs. Forelimb (FL) and hind limb (HL) DNaseI hypersensitivity tracks from ENCODE are shown (*Rosenbloom et al., 2013*), along with vertebrate sequence conservation (*Blanchette et al., 2004*). (**B-D**) Whole-mount lacZ staining of transgenic embryos carrying *PelB* enhancer constructs, with close ups of the limbs. Dotted lines denote boundaries of limbs.

DOI: https://doi.org/10.7554/eLife.38555.003

The following figure supplements are available for figure 2:

**Figure supplement 1.** Comparison of *Pitx1* locus H3K27ac profile in mouse and Anolis lizard embryonic forelimbs and hind limbs.

DOI: https://doi.org/10.7554/eLife.38555.004

**Figure supplement 2.** *PelB* conservation in vertebrates.

DOI: https://doi.org/10.7554/eLife.38555.005

*Pitx1*[PelB-/PelB-] animals as compared to wild type. However, significant quantitative reductions were seen in the length of several hind limb bones in the homozygous *PelB* mutant mice, confirming that the *PelB* enhancer sequence is required for normal developmental size of hind limb structures (*Figure 3*, *Supplementary file 1D*).

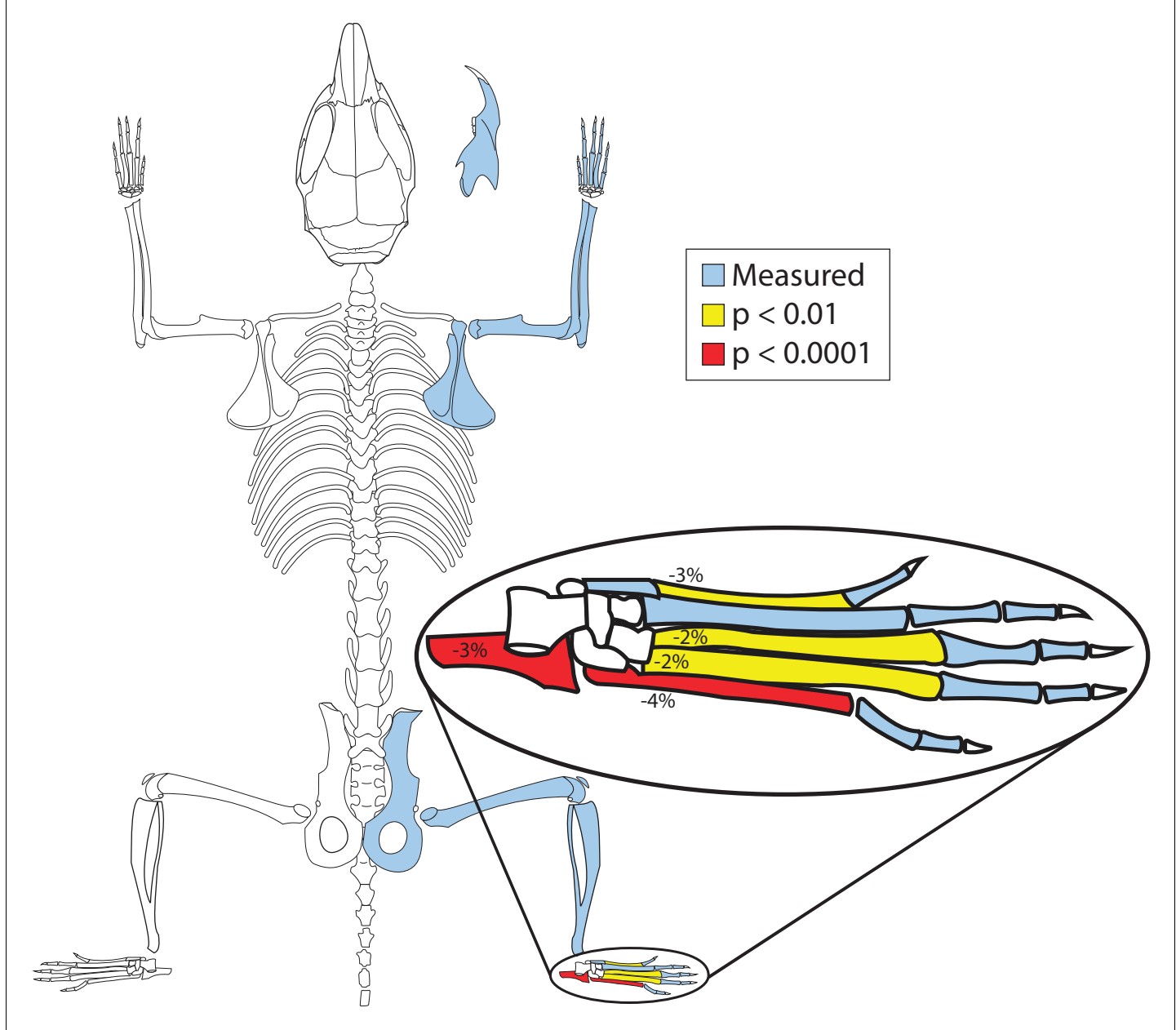

**Figure 3.** *PelB* deletion mice have smaller hind feet. Bones with significant length changes between control (*Pitx1*$^{+/+}$) and enhancer knockout (*Pitx1*$^{PelB-/PelB-}$) mice are highlighted (Blue = measured, Yellow = $p < 0.01$, Red = $p < 0.0001$). Numbers indicate percent change in mean length. (*Pitx1*$^{+/+}$ n=21, *Pitx1*$^{PelB-/PelB-}$ n = 20).

DOI: https://doi.org/10.7554/eLife.38555.006

To test whether *PelB* deletion has additional phenotypic effects in a sensitized background with lower levels of *Pitx1*, we crossed *Pitx1*$^{PelB-/PelB-}$ animals to mice heterozygous for a functional knockout mutation that disrupts the protein-coding region of the *Pitx1* gene (*Pitx1*$^{null/+}$) (*Szeto et al., 1999*). *Pitx1*$^{null/null}$ mice show severe hind limb deformities, facial abnormalities, and neonatal lethality (*Lanctôt et al., 1999*; *Szeto et al., 1999*). By contrast, heterozygous *Pitx1*$^{null/+}$ mice are reported to be phenotypically normal, with a small percentage exhibiting a club foot phenotype (*Alvarado et al., 2011*). We compared *Pitx1*$^{null/PelB-}$ mice to *Pitx1*$^{null/+}$ to determine if loss of the *PelB* sequence alters hind limb development.

All heterozygous carriers of the *Pitx1* null mutation were smaller than non-carriers, with significant reduction of jaw and several hind limb elements after correcting for body size (*Supplementary file 1E*). In addition, *Pitx1*$^{null/PelB-}$ mice showed significant reductions in length of the jaw and several hind limb bones compared to *Pitx1*$^{null/+}$ animals (*Supplementary file 1F*). The most striking phenotype was a complete absence of patellae in *Pitx1*$^{null/PelB-}$ mice (*Figure 4K,L*). For comparison, patellae are completely missing in *Pitx1*$^{null/null}$ mice (*Lanctôt et al., 1999*; *Szeto et al., 1999*) and are smaller but consistently present in the *Pitx1*$^{null/+}$ heterozygous animals in our crosses (*Figure 4H,I*). Patellae are found in hind limbs but not forelimbs of many vertebrates (*Samuels et al., 2017*), and loss of the patella in the compound crosses suggests that *PelB* also contributes to the development of this structure.

### *PelB* enhancer activity is conserved from mouse to fish

Because of the high sequence conservation of *PelB* across vertebrates, we tested whether the orthologous sequence from teleosts can also function as an enhancer in developing pelvic appendages. Stickleback *PelB* shows 51% identity to mouse *PelB* over a core 530 bp region. We cloned the *PelB* sequence from a marine stickleback population (Salmon River *PelB*; SALR-*PelB*) upstream of a GFP reporter, and injected the reporter construct into fertilized eggs of pelvic-complete sticklebacks. Transient transgenic larvae are readily identifiable by GFP expression driven by the reporter vector itself in the lens of the eye (*Nagayoshi et al., 2008*). The SALR-*PelB* constructs drove consistent GFP expression in the developing pelvic spines and girdle (*Figure 5B*). Interestingly, very strong GFP was also observed in the developing jaw region, another site of normal *Pitx1* expression (*Shapiro et al., 2004*). We note that the 530 bp region contains many predicted transcription factor binding sites, including a putative *Pitx1* binding domain (*Figure 5—figure supplement 1–3*), which may contribute to the tissue-specific pattern.

### *PelB* enhancer is disrupted in a pelvic-reduced stickleback population

To determine whether *PelB* sequences are modified in wild populations with evolutionary changes in pelvic hind fin development, we sequenced *PelB* from multiple freshwater stickleback populations that show loss or reduction of pelvic structures. We observed multiple sequence variants and small indels in the region, most of which did not correlate with pelvic status (*Figure 5—figure supplement 1–3*). However, one stickleback population with extreme pelvic reduction (Paxton Lake benthic, PAXB) showed a large compound indel in the *PelB* region (*Figure 5*, *Figure 5—figure supplement 1–3* for full sequence). The PAXB population is interesting, as it exhibits one of the most extreme examples of pelvic loss, with most PAXB fish showing no pelvic bones at all (*McPhail, 1992*), and also has a known deletion in the previously identified *Pel* region located upstream of *Pitx1* (*Chan et al., 2010*), which we now refer to as *PelA*. PAXB fish have a deletion of 125 bp and an insertion of 341 bp in the *PelB* sequence. The inserted sequence is nearly identical to a region located approximately 3 kb upstream of the *PelB* region, suggesting the allele arose by concurrent duplication and local insertion of DNA.

To determine whether this indel mutation alters the function of *PelB*, we tested the enhancer activity of PAXB-*PelB* in pelvic-complete stickleback embryos in the GFP reporter assay. As with the marine construct, PAXB-*PelB* drove very strong, consistent reporter expression in the eye and jaws of transgenic fish (*Figure 5C,D*, *Figure 5—figure supplement 4*). However, the PAXB-*PelB* construct drove weak and limited expression in the developing pelvic region compared to controls, a difference that we confirmed by quantitative measurement of the extent of GFP expression in many independent transgenic larvae (*Figure 5C,E*). Interestingly, PAXB-*PelB* also exhibited novel expression in the developing pectoral fin rays, a pattern not observed with the marine construct. The novel deletion and insertion of sequence in the PAXB-*PelB* region thus leads to both reduction of enhancer activity in the pelvis, and gain of new activity in other fin rays.

## Discussion

Despite the key role of the *Pitx1* gene in specification of hind limb identity and development, the sequences controlling its normal expression pattern are still poorly understood. While an upstream pelvic-specific enhancer (*PelA*) was previously identified in sticklebacks (*Chan et al., 2010*), this sequence is not well conserved outside teleosts. Regulatory mutations in birds, humans, and mice

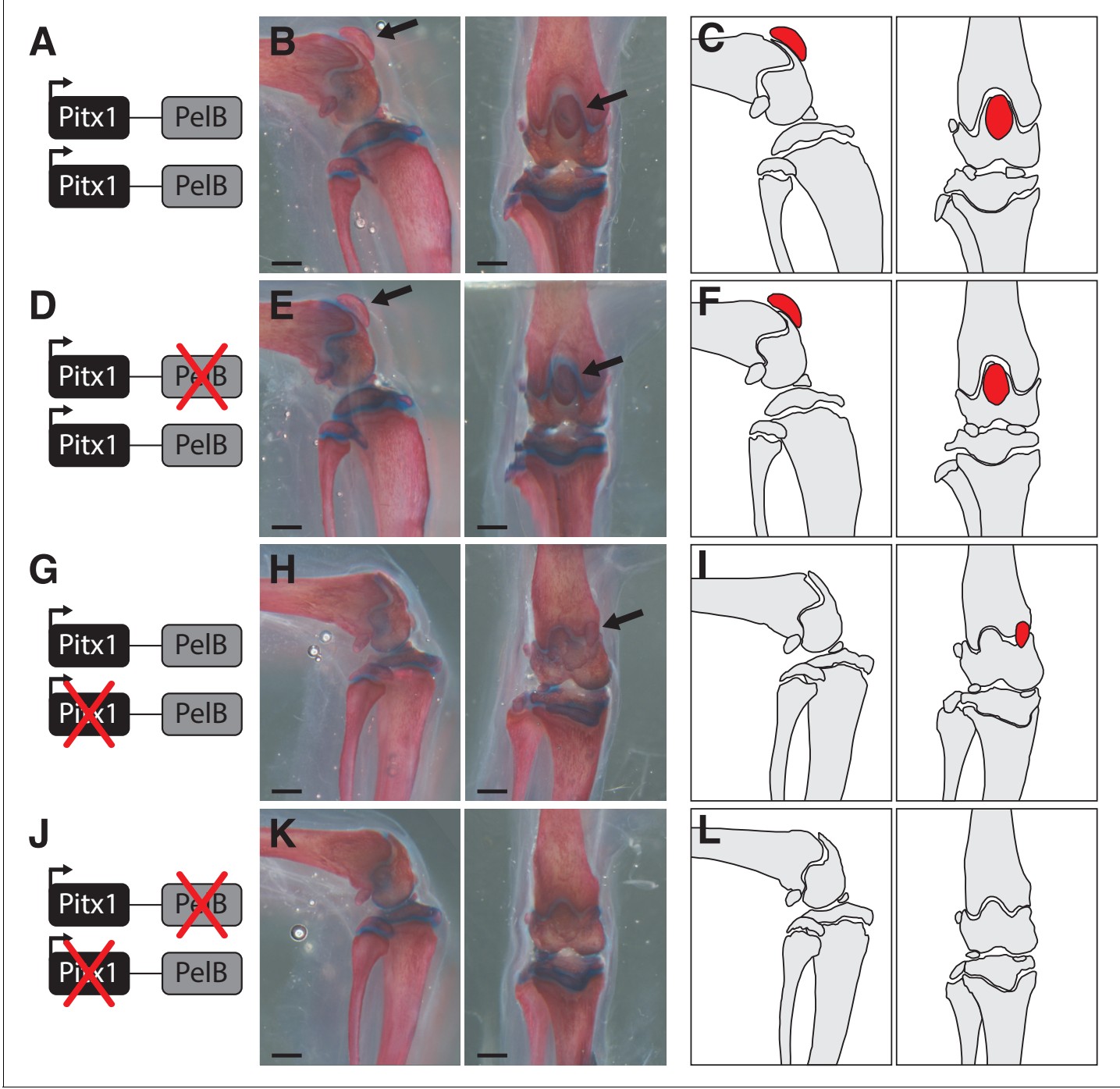

**Figure 4.** Loss of patellae in mice missing the *PelB* enhancer and one functional copy of *Pitx1*. (A,D,G,J) Gene diagrams showing the status of the *PelB* enhancer and *Pitx1* coding region produced by crossing *PelB* knockout and *Pitx1* null mutant lines. (B,E,H,K) Mouse knee joint, with patellae indicated with a black arrow. Scale bars = 1 mm. (C,F,I,L) Outlines of bones, with patellae highlighted in red.

DOI: https://doi.org/10.7554/eLife.38555.007

implicate multiple 5' upstream regions in *Pitx1* regulation (*Spielmann et al., 2012*; *Domyan et al., 2016*; *Kragesteen et al., 2018*), and several enhancers have now been identified in the upstream regions that are well conserved in tetrapods but not fish (*Spielmann et al., 2012*; *Sarro et al., 2018*; *Kragesteen et al., 2018*). Although multiple sequences in the mouse 5' region clearly contribute to overall *Pitx1* expression, no individual enhancer from the upstream region can recapitulate hind

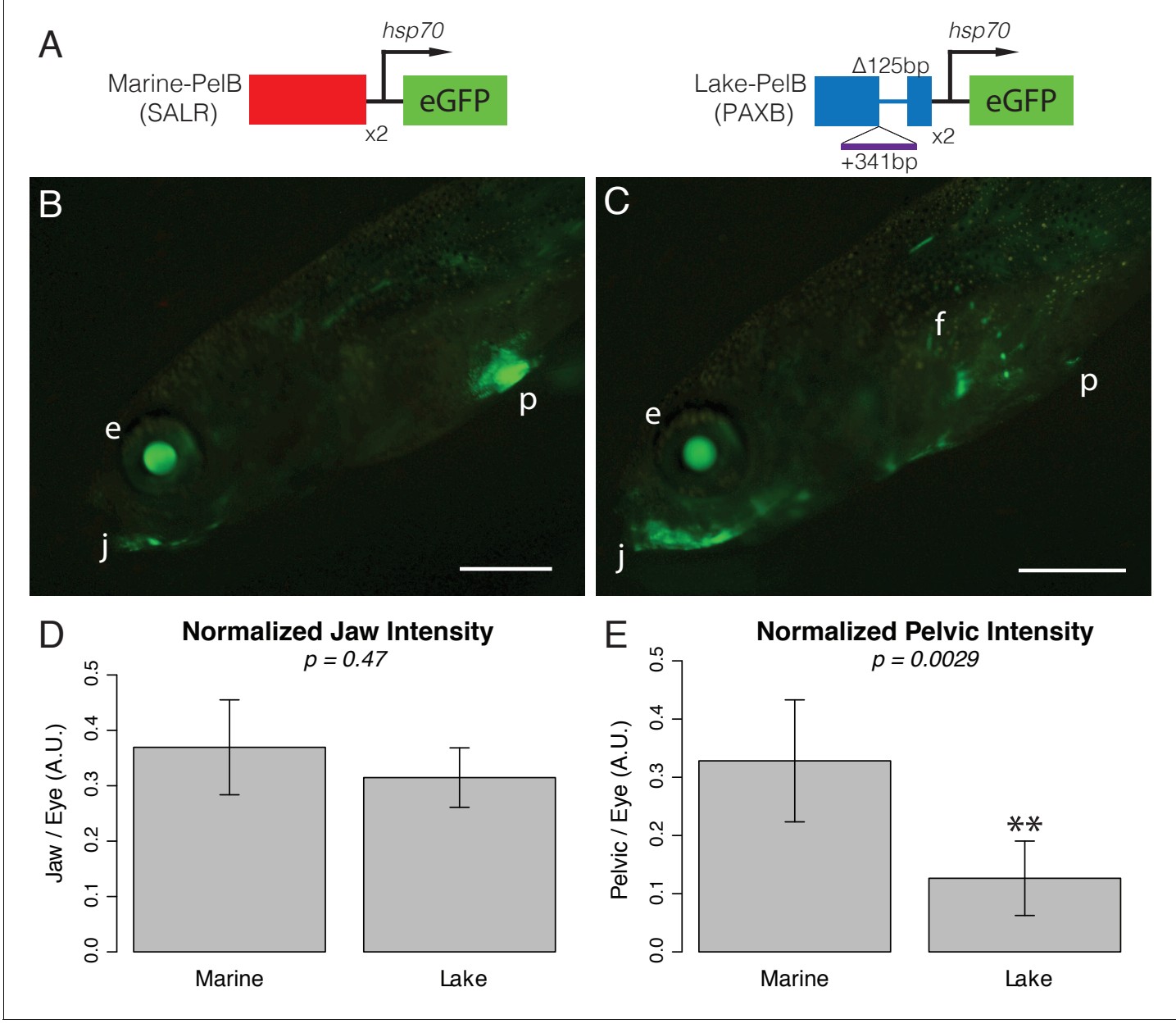

**Figure 5.** *PelB* enhancer function is conserved to fish. (A) Schematic of enhancer constructs tested in stickleback fish. Each enhancer was cloned as a 2x concatemer upstream of an hsp70 promoter and eGFP. The marine sequence is from the Salmon River (SALR) population, and the lake sequence is from Paxton benthic (PAXB). (B-C) GFP fluorescence in live sticklebacks. GFP expression in the lens of the eye (e) is driven by the hsp70 promoter itself, allowing easy identification of transgenic fish. (B) The marine *PelB* enhancer drives expression in the jaw (j) and pelvis (p). (C) The lake *PelB* enhancer drives similar expression in the jaw (j), but significantly less expression in the pelvis (p), as well as a novel expression in the fin rays (f). (D-E) Quantified fluorescence in GFP-positive fish that show pelvic expression, normalized to intensity in the lens of the eye to control for copy number and integration site. SALR n = 24, PAXB n = 18. Scale bars = 1 mm.

DOI: https://doi.org/10.7554/eLife.38555.008

The following figure supplements are available for figure 5:

**Figure supplement 1.** *PelB* sequence variation in stickleback fish (1-300).
DOI: https://doi.org/10.7554/eLife.38555.009

**Figure supplement 2.** *PelB* sequence variation in stickleback fish (301-600).
DOI: https://doi.org/10.7554/eLife.38555.010

**Figure supplement 3.** *PelB* sequence variation in stickleback fish (601-895).
DOI: https://doi.org/10.7554/eLife.38555.011

*Figure 5 continued on next page*

*Figure 5 continued*

**Figure supplement 4.** Quantified *PelB* enhancer activity in stickleback fish.

DOI: https://doi.org/10.7554/eLife.38555.012

limb-specific expression, and knockouts of single upstream enhancers lead to incomplete reduction of hind limb expression, and mild or undetectable hind limb phenotypes (*Kragesteen et al., 2018*; *Sarro et al., 2018*). We have now identified a novel enhancer from the mammalian *Pitx1* gene that is located downstream rather than upstream of *Pitx1* coding exons, in a region that is also known to interact directly with the *Pitx1* promoter during hind limb development (*Andrey et al., 2017*; *Kragesteen et al., 2018*; *Supplementary file 1B*). This is the first mammalian enhancer from the *Pitx1* region that can drive hind limb-specific rather than pan-limb expression. *PelB* is also highly conserved between mice and sticklebacks, suggesting that ancient genomic mechanisms controlling expression of *Pitx1* in pelvic appendages already existed in the common ancestors of tetrapods and fish.

Although the *PelB* enhancer shows strong hind limb specific expression in the developing mouse embryo, the expression pattern is restricted to the posterior side of the autopod. By contrast, the endogenous *Pitx1* gene is expressed throughout the developing hind limb (*Szeto et al., 1999*; *Sarro et al., 2018*), suggesting that other control sequences also must contribute to *Pitx1* regulation. Knockout of *PelB* leads to only partial reduction of *Pitx1* expression in hind limbs, as expected if other enhancers are also involved in *Pitx1* regulation. Recent studies show that deletion of 5' regulatory regions can also lead to reproducible but incomplete reduction of *Pitx1* expression in hind limbs, showing that multiple control sequences located both downstream and upstream of *Pitx1* are required for normal expression (*Kragesteen et al., 2018*; *Sarro et al., 2018*). Our phenotypic studies in mice show that loss of the *PelB* enhancer also leads to milder hind limb phenotypes than inactivation of the *Pitx1* gene itself. However, loss of the *PelB* enhancer alone significantly reduces the length of several hind limb bones, and loss on a sensitized background with a heterozygous *Pitx1* null mutation leads to complete loss of the patella. These studies confirm that the *PelB* enhancer is required for normal pelvic hind limb development in mammals, but has larger phenotypic effects on some hind limb structures than others.

The patella is a large sesamoid bone embedded in the tendon connecting the femur and tibia (*Samuels et al., 2017*). Embryological studies show that the patella arises from dorsal limb bud mesenchyme cells and requires muscular contraction for normal development (*Eyal et al., 2015*). It is possible that *PelB* expressing cells in the posterior limb bud contribute descendants to the dorsal limb bud domain that forms the patella, or to domains that form more anterior skeletal structures. For comparison, previous fate mapping experiments have shown that the descendants of *Shh* expressing cells in the posterior limb bud expand and contribute to multiple skeletal structures outside the initial expression domain (*Harfe et al., 2004*). Alternatively, loss of the *PelB* enhancer could have indirect effects on either the patella or more anterior skeletal structures, for example by altering overall chromatin configuration of the *Pitx1* locus (*Kragesteen et al., 2018*) and thus changing additional *Pitx1* expression domains, by perturbing cell non-autonomous signaling pathways involved in limb formation, or by altering patterns of muscular activity that also influence formation of some skeletal structures. In the future, these possibilities can be further tested by mapping the developmental fate of *PelB* expressing cells in limb buds, by using chromosome conformation approaches to study the configuration of the *Pitx1* locus in mice missing the *PelB* enhancer, or by looking for additional expression and anatomical changes in *PelB* mutants, including possible changes in non-skeletal tissues like muscle.

Recent studies (*Osterwalder et al., 2018*) have proposed that many vertebrate genes have multiple 'functionally redundant' enhancers that drive similar patterns in a particular tissue, with each enhancer making purely quantitative contributions to the overall level of expression of a gene. Some of our results superficially appear to fit this model, including the existence of multiple enhancers that contribute to *Pitx1* expression in developing pelvic appendages (such as the two separate pelvic control regions, *PelA* and *PelB*, now identified in the fish *Pitx1* gene), and obvious presence/absence phenotypes that are seen only when enhancer knockouts are analyzed on sensitized rather than wild type backgrounds (such as complete loss of the patella seen in $Pitx1^{null/PelB-}$ but not $Pitx1^{PelB-/PelB-}$

mice). At the same time, our studies provide multiple lines of evidence indicating that *Pitx1* enhancers are not truly redundant.

First, while *PelB* is active in the hind limbs of mice, it is clearly not active throughout the entire hind limb, as might be expected if it was making a purely quantitative contribution to the overall *Pitx1* hind limb pattern. Similar results have been seen for other genes with a key role in hind limb or skeletal development. For example, two different hind limb enhancers have previously been identified in the *Tbx4* gene of mammals (*HLEA* and *HLEB*). Although these two enhancers drive expression patterns that overlap in some regions of the hind limb, each enhancer also has unique spatial expression patterns not shared with the other enhancer (*Menke et al., 2008*; *Infante et al., 2015*).

Second, significant phenotypes are seen in laboratory animals engineered to lack individual enhancers, a result that clearly demonstrates the enhancers are not functionally redundant at the organismal level. These phenotypes may be quantitative rather than qualitative (such as length changes rather than presence or absence of metatarsals), or restricted to particular subsets of anatomy, or only revealed under particular environmental conditions (*Frankel et al., 2010*). We note that similar results have also been reported for the two different hind limb enhancers of the *Tbx4* gene. Knockout of *HLEA* alone produces significant changes in length of hind limb elements, even if additional, or more severe, phenotypes are detected when analyzed on a genetic background heterozygous for a *Tbx4* null mutation (*Menke et al., 2008*; *Infante et al., 2015*).

Third, mutations of individual *Pitx1* enhancers have clearly made significant contributions to adaptive evolution of skeletal phenotypes in wild populations. Many pelvic-reduced sticklebacks show complete deletion of the *PelA* but not the *PelB* enhancer in natural environments, as well as molecular signatures of adaptive selection that are centered on the *PelA* region (*Chan et al., 2010*). These results suggest that loss of the *PelA* enhancer alone is sufficient to produce evolutionarily significant pelvic reduction in some wild populations.

Because the *PelA* enhancer of sticklebacks is alignable in fish but not tetrapods, while the *Pen* enhancer is found in tetrapods but not fish, it has recently been proposed that hind limb specific regulation of *Pitx1* may have arisen by different mechanisms in fish and tetrapod lineages (*Kragesteen et al., 2018*). Within tetrapods, the *Pen* enhancer also drives expression in both forelimbs and hind limbs unless silenced by other elements, raising the possibility that the ancestral form of *Pitx1* expression was actually pan-limb rather than hind limb-specific, and that distinct *Pitx1* expression patterns and distinct morphology of forelimbs and hind limbs evolved as subsequent events in the tetrapod lineage (*Kragesteen et al., 2018*). In contrast, our data identify a *Pitx1* enhancer whose sequence is clearly conserved between fish and mammals, and that also drives expression in pelvic rather than pectoral appendages when functionally tested in both sticklebacks and mice. These data show that pelvic-appendage-specific enhancers of the *Pitx1* gene are a shared and ancient genomic feature that already existed in the common ancestor of both fish and land animals. An ancient and ancestral *PelB*-based regulatory mechanism may have subsequently been modified by evolution of separate lineage-specific *PelA* and *Pen*-based regulatory sequences in teleost fish and tetrapods. Alternatively, *PelA*-like and *Pen*-like functions may also be ancient like *PelB*, even though the primary sequence of these enhancers are no longer alignable because of turnover of binding sites at the DNA sequence level (*Dermitzakis and Clark, 2002*; *Wray et al., 2003*). We note that the existence of ancient shared enhancers in limb identity genes like *Pitx1* is consistent with the fossil record of fin and limb evolution. Genetic programs for producing morphologically distinct pectoral and pelvic appendages have clearly existed since early periods of vertebrate evolution, as shown by the distinct morphologies of pectoral and pelvic fins in stem gnathosomes, as well as the distinct forelimb and hind limb morphologies of stem tetrapods (*Coates, 2003*; *Zhu et al., 2012*; *Shubin et al., 2014*; *Jeffery et al., 2018*).

Identification of the evolutionarily conserved *PelB* hind limb enhancer in the *Pitx1* gene will facilitate additional studies of limb development in a wide range of vertebrate species. For example, hind limb and patellar abnormalities are relatively common in humans (*Bongers et al., 2005*; *Zhang et al., 2017*; *Basit and Khoshhal, 2018*). The sequence of the *PelB* region can now be examined for base pair changes, insertions, deletions, or copy number differences that may contribute to morphological traits or disease phenotypes in humans.

Our data in sticklebacks already provide one clear example of *PelB* sequence changes that are linked to dramatic changes in pelvic structures in natural populations. Benthic sticklebacks from Paxton Lake in British Columbia show almost complete loss of the pelvic apparatus, one of the key

skeletal differences that distinguish the classic benthic-limnetic species pair found within this lake (*McPhail, 1992*). In previous genetic crosses with Paxton benthic fish, most of the variation in pelvic size has been linked to a 300 kb region surrounding the *Pitx1* locus (*Shapiro et al., 2004*; *Chan et al., 2010*). Here we show that PAXB fish have dual mutations in two different pelvic enhancers of the *Pitx1* gene. While the *PelA* enhancer is completely deleted (*Chan et al., 2010*), the *PelB* enhancer shows both a 125 bp deletion and a 341 bp insertion in PAXB fish. We have confirmed that these large structural changes reduce the hind limb enhancer activity of the *PelB* enhancer in transgenic reporter assays, and also generate a novel expression pattern in developing rays of the pectoral and median fins. Paxton benthic fish do not show distinctive morphologies in their pectoral and median fins compared to other freshwater sticklebacks, though it is possible that subtle morphological changes have been missed. In contrast, the dual mutations in both *PelA* and *PelB* in Paxton benthic fish likely contribute to the nearly complete loss of the pelvic apparatus in this population, compared to the partial pelvic reduction observed in many populations missing only the *PelA* enhancer. In the future, it will be interesting to engineer fish that carry only the *PelA* deletion, only the *PelB* alteration, or alterations in both sequences upstream and downstream of the *Pitx1* locus. This will make it possible to reconstruct the phenotypic contributions that each enhancer makes to different structures within the pelvic fin, and to the overall size of pelvic appendages in a fish model of skeletal evolution.

Many other vertebrates show interesting skeletal changes in hind limbs, forelimbs, or both. Initial examination shows that the primary sequence of *PelB* is largely intact in whales and manatees (*Figure 2—figure supplement 2*). As minor sequence alterations in transcription factor binding sites can dramatically alter enhancer functionality (*Lettice et al., 2003*; *Lettice et al., 2008*; *Kvon et al., 2016*), further studies will be required to see if any individual base pair changes have altered *PelB* function in these two different mammalian clades that show dramatic hind limb reduction. We note that multiple advanced snake species appear to have deleted a portion of the *PelB* sequence (*Figure 2—figure supplement 2*). Interestingly, the corresponding sequence is still present in pythons, a snake that retains small external hind limbs (*Leal and Cohn, 2016*). Identification of *PelB* and other enhancers of the *Pitx1* gene should make it possible to test whether particular *cis*-acting regulatory alterations have contributed to evolutionary changes in hind limb morphology in these and other vertebrate species.

## Materials and methods

**Key resources table**

| Reagent type (species) or resource | Designation | Source or reference | Identifiers | Additional information |
|---|---|---|---|---|
| Gene (*Mus musculus*) | *Pitx1* | NA | RefSeq: NM_011097.2 | |
| Gene (*Mus musculus*) | *Pcdb2* | NA | RefSeq: NM_028281.1 | |
| Gene (*Gasterosteus aculeatus*) | *Pitx1* | NA | NCBI GU130437.1, GU130434.1 | |
| Strain, strain background (*G. aculeatus*) | Salmon River marine population | DOI: 10.1126/science.1182213 | SALR, GU130434.1 | Migratory marine fish with complete pelvis |
| Strain, strain background (*G. aculeatus*) | Paxton Lake benthic freshwater population | DOI: 10.1126/science.1182213 | PAXB, GU130437.1 | Lake-resident fish with no pelvis |
| Strain, strain background (*G. aculeatus*) | Matadero Creek freshwater population | DOI: 10.1126/science.1182213 | MATA | Stream-resident fish with complete pelvis |
| Genetic reagent (*M. musculus*) | *Pitx1* coding region mutation | DOI: 10.1101/gad.13.4.484. | *Pitx1$^{null}$* | Knock out generated in M. Rosenfeld lab; transferred to C. Gurnett, then D. Menke, Kingsley lab. |

*Continued on next page*

*Continued*

| Reagent type (species) or resource | Designation | Source or reference | Identifiers | Additional information |
|---|---|---|---|---|
| Genetic reagent (*M. musculus*) | *Pitx1 PelB* deletion mutation | this paper | *Pitx1^PelB-* | |
| Transfected construct (*M. musculus*) | RP23-55J18 BAC clone | bacpacresources.org/femmouse23.htm | | |
| Transfected construct (*M. musculus*) | RP24-296J14 BAC clone | bacpacresources.org/mmouse24.htm | | |
| Transfected construct (*M. musculus*) | RP24-353M13 BAC clone | bacpacresources.org/mmouse24.htm | | |
| Transfected construct (*M. musculus*) | RP23-231P13 BAC clone | bacpacresources.org/femmouse23.htm | | |
| Transfected construct (*M. musculus*) | RP23-233P6 BAC clone | bacpacresources.org/femmouse23.htm | | |
| Transfected construct (*M. musculus*) | PelB-lacZ | this paper | pTCPcbd2-1 | |
| Transfected construct (*M. musculus*) | PelBcon1-lacZ | this paper | pTCPcbd2-4 | |
| Transfected construct (*M. musculus*) | PelBcon2-lacZ | this paper | pTCPcbd2-2 | |
| Transfected construct (*G. aculeatus*) | SALR-PelB-GFP | this paper | SALR-PelB | |
| Transfected construct (*G. aculeatus*) | PAXB-PelB-GFP | this paper | PAXB-PelB | |
| Recombinant DNA reagent | basal promoter lacZ vector | DOI: 10.1073/pnas.97.4.1612 | hspLacZ | |
| Recombinant DNA reagent | basal promoter GFP vector | DOI: 10.1186/gb-2007–8 s1-s7 | pT2HE | |
| Sequence-based reagent (*M. musculus*) | ENCODE DNaseI hypersensitivity peaks in developing forelimbs and hind limbs | DOI: 10.1093/nar/gks1172 | | |
| Sequence-based reagent (*M. musculus*) | Pitx1 promoter-interacting regions with histone marks of enhancers | DOI: 10.1101/gr.213066.116 | | |
| Sequence-based reagent (*M. musculus*) | H3K27ac peaks in developing forelimbs and hind limbs | DOI: 10.1016/j.devcel.2015.09.003 | | |
| Sequence-based reagent (*Anolis carolinensis*) | H3K27ac peaks in developing forelimbs and hind limbs | DOI: 10.1016/j.devcel.2015.09.003 | | |

## Comparative sequence analysis

Sequences from *Mus musculus, Homo sapiens, Bos taurus, Tursiops truncatus, Loxodonta africana, Trichechus manatus, Anolis carolinensis, Python bivittatus, Boa constrictor, Pantherophis guttatus, Thamnophis sirtalis, Ophiophagus hannah, Vipera berus, Protobothrops mucrosquamatus, Xenopus laevis, Lepisosteus oculatus,* and *Gasterosteus aculeatus* (Salmon River, clone BAC CHORI213-118G22, GU130434.1; Paxton Lake benthic, clone BAC CHORI215-196J14, GU130437.1) were

obtained from NCBI or Assemblathon2 (*Boa constrictor*, *Bradnam et al., 2013*), aligned with Shuf-fle-LAGAN and analyzed with VISTA (*Brudno et al., 2003*; *Frazer et al., 2004*).

## BAC scan and mouse transgenic reporter assays

Each BAC clone was co-injected with a minimal promoter-lacZ reporter construct to generate transient transgenic embryos as described (*DiLeone et al., 2000*). Potential enhancer regions were amplified using primers (*Supplementary file 1G*) containing NotI restriction sites and cloned into the NotI site of p5-Not-Hsp68LacZ (*DiLeone et al., 1998*). The resulting expression constructs contain the following mouse genomic regions: *PelB* (pTCPcbd2-1), mm9 chr13: 55,850,296–55,859,761; *PelBcon1* (pTCPcbd2-4), mm9 chr13: 55,850,593–55,852,765; and *PelBcon2* (pTCPcbd2-2), mm9 chr13: 55,854,269–55,857,548. Prior to microinjection, plasmid DNAs were purified as described (*DiLeone et al., 2000*). Pronuclear injection into FVB embryos was performed by Taconic Biosciences and Cyagen Biosciences. Microinjected embryos were collected at E12.5 and stained with X-gal as described (*DiLeone et al., 1998*).

## Generation of *PelB* enhancer-knockout mice

For CRISPR-Cas9 targeting of the *PelB* locus, potential guide RNAs were screened in cell culture lines for efficacy. Pronuclear injection of the top 4 gRNAs (*Supplementary file 1G*) into C57BL/6J embryos was performed to generate transient transgenic $F_0$ mice (HHMI Janelia Farms). Eighteen independent tail positive founder mice were bred to produce stable lines. A line containing a clean deletion with breakpoints closely matching the boundaries of the *PelB* enhancer construct was saved for further analysis. The deleted region in this line corresponds to mm9 chr13:55,850,296–55,859,733. For comparison, the region cloned for the full-length *PelB* enhancer construct corresponds to mm9 chr13: 55,850,296–55,859,761.

## qRT-PCR

Left hind limbs were collected from E12.5 embryos and stored in RNAlater (Thermo Fisher Scientific). RNA was prepared using the RNeasy Mini kit (Qiagen) with the on-column DNaseI digestion step. 400 ng of RNA was treated with an additional DNaseI digestion step (Invitrogen) before cDNA was prepared using SuperScript III First Strand synthesis kit (Invitrogen). qRT-PCR was performed using a 1:20 dilution of each cDNA with primers described in *Supplementary file 1G* and 2X Brilliant II SYBR QPCR Low ROX Master Mix (Agilent) on a QuantStudio five system (ThermoFisher) using the Standard Curve with Melt 2-step program. All reactions were done in triplicate within an experimental run, and the average of the three values was used for further analysis. Levels of gene expression were determined using standard curves for each primer set, constructed using known amounts of unrelated E12.5 limb cDNA as templates. All mutant and wild type samples were assayed together, and standard curves were performed in each assay to control for plate-to-plate variation. The relative levels of *Pitx1* and *Pcbd2* gene expression in a sample were normalized to the corresponding level of expression of the reference gene, *Pgk1*, in the same sample.

## Mouse skeletal preparations

Male mice were collected at approximately P28. Mice were weighed, skinned, eviscerated, placed in PBS, and then fixed in 95% ethanol for at least 48 hr, with the solution replaced daily. Skeletons were incubated in 760 mL 95% ethanol + 200 mL glacial acetic acid + 50 mL alcian blue solution (2% alcian blue in 70% ethanol). After 6 days of staining, skeletons were rinsed in 95% ethanol for 48 hr, with the solution replaced daily. Skeletons were then placed in 0.8% KOH for 4 days, followed by 1% KOH + 0.00015% alizarin red for 48 hr. Skeletons were then placed in 50% glycerol overnight, followed by 100% glycerol overnight. Skeletons were then transferred to 100% glycerol +thymol crystals for storage. All incubations were done on an orbital shaker for even staining.

## Mouse morphological measurements

Mouse skeletal preparations were disarticulated under a dissecting microscope to remove the right jaw, forelimb, pelvis, and hind limb (n = 21 *Pitx1*$^{+/+}$ mice, n = 20 *Pitx1*$^{PelB-/PelB-}$ mice; n = 20 *Pitx1*$^{+/+}$ mice, n = 25 *Pitx1*$^{+/PelB-}$ mice, n = 22 *Pitx1*$^{null/+}$ mice, n = 22 *Pitx1*$^{null/PelB-}$ mice). Bones were flattened using a coverslip, photographed, and measured in FIJI (*Schindelin et al., 2012*). As all bone

length traits are correlated with mouse size, residuals from a linear regression on humerus length were used for all traits. Residuals were compared using a two-sided T-test. For patella presence vs absence, a Fisher's Exact test was used. Mice exhibiting polydactyly were excluded from comparison, and blinding was not part of the study design.

## Stickleback crosses and husbandry

Lab-reared fish were raised in 30-gallon tanks under common conditions (2.8 g/L Instant Ocean salt) and fed live brine shrimp as larvae, then frozen daphnia, bloodworms, and mysis shrimp as juveniles and adults. Pelvic development is visible starting when the fish are 7.5 mm standard length, and is considered to be complete by the time the fish reach 16.5 mm (*Bell and Harris, 1985*).

## Stickleback *PelB* enhancer

The stickleback pelvic enhancer region was amplified from BAC DNA from either marine fish from Salmon River (clone CHORI213-118G22, NCBI Genbank accession GU130434.1) or Paxton Lake benthic (clone CHORI215-196J14, GU130437.1) using primers with NheI and AvrII recognition sites (*Supplementary file 1G*). To make directional concatemers, PCR fragments were digested with NheI and AvrII, ligated together, and blunted with Klenow. 2x concatemers were gel extracted and cloned in to the EcoRV site of the pT2HE GFP reporter vector (modified from *Kawakami, 2007*) to generate either SALR-*PelB* or PAXB-*PelB*.

## Transgenic stickleback assays

Transgenic sticklebacks were generated by microinjection of freshly fertilized eggs as previously described (*Chan et al., 2010*). Plasmids were co-injected with Tol2 transposase mRNA as described (*Hosemann et al., 2004*). Mature Tol2 mRNA was synthesized by *in vitro* transcription using the mMessage mMachine SP6 kit (Life Technologies). All enhancer assays were performed on pelvic-complete stickleback from Matadero Creek, California, USA (MATA). All larvae were raised under standard aquarium conditions to Swarup St 29/30 (*Swarup, 1958*), when pelvic bud development is initiated, for phenotyping. Larvae were anesthetized in 0.0003% w/v tricaine (Ethyl 3-aminobenzoate methanesulfonate, Sigma). Microscopic observation for GFP expression was conducted with a MZFLIII fluorescent microscope (Leica Microsystems, Bannockburn, IL) using GFP2 filters and a ProgResCF camera (Jenoptik AG, Jena, Germany) to distinguish GFP expression from autofluorescence in pigmented fish.

## GFP quantification

GFP intensity in 24 and 18 independent transgenic larvae carrying Salmon River or Paxton Benthic transgenes respectively was quantified in FIJI (*Schindelin et al., 2012*). In the green channel, a circle was drawn in the lens of the eye, and the mean intensity was measured. The same circle was placed over the pelvis to calculate mean intensity, and then placed on an adjacent non-GFP region to calculate background fluorescence; the background value was subtracted from the raw pelvic value to generate a pelvic GFP score. Jaw intensity was calculated by measuring mean intensity of a line drawn through the jaw, and an adjacent non-GFP region was measured with the same line for background fluorescence; the background value was subtracted from the raw jaw value to give jaw GFP score. Pelvic and jaw scores were normalized to eye intensity to control for construct integration location and copy number and compared with a Mann-Whitney U Test (*Figure 5—figure supplement 4*).

## Primers

All primers used in construct design, genotyping, sequencing, and qRT-PCR are listed in *Supplementary file 1G*.

## Acknowledgements

We thank C Guo for CRISPR/Cas9 targeting of the *PelB* region in mice; M Rosenfeld and C Gurnett for null mice; M A Bell and A MacColl for stickleback samples; M A Bell, K Xie, G Kingman, and J Wucherpfennig for assistance in fieldwork; S Brady and E Au for fishroom assistance; and C Gurnett,

M Fuller, G Sherlock, D Vollrath, M A Bell, C Lowe, J Wucherpfennig, H Chen, A Kessler, and all members of the Kingsley lab for helpful discussions. This work was supported in part by the Stanford Genome Training Program (5T32HG000044, ACT), a Ruth L Kirschstein National Research Service Award from the National Institute of Arthritis and Musculoskeletal and Skin Diseases (1F31AR068870, ACT), a National Science Foundation CAREER award (IOS-1149453, DBM), the National Institute of Child Health and Human Development (HD081034, DBM), and a NIH Center of Excellence in Genomic Science grant (NHGRI 3P50 HG002568, DMK). DMK is an investigator of the Howard Hughes Medical Institute.

## Additional information

### Funding

| Funder | Grant reference number | Author |
|---|---|---|
| National Human Genome Research Institute | Stanford Genome Training Program - 5T32HG0004 | Abbey C Thompson |
| National Institute of Arthritis and Musculoskeletal and Skin Diseases | Ruth L. Kirschstein National Research Service Award - 1F31AR068870 | Abbey C Thompson |
| National Science Foundation | CAREER Award - IOS-1149453 | Douglas B Menke |
| National Institute of Child Health and Human Development | HD081034 | Douglas B Menke |
| National Human Genome Research Institute | Center of Excellence in Genomic Science - 3P50 HG002568 | David M Kingsley |
| Howard Hughes Medical Institute | | David M Kingsley |

The funders had no role in study design, data collection and interpretation, or the decision to submit the work for publication.

### Author contributions

Abbey C Thompson, Conceptualization, Data curation, Formal analysis, Funding acquisition, Validation, Investigation, Visualization, Methodology, Acquisition of data, Analysis and interpretation of data, Writing—original draft, Writing—reviewing and editing; Terence D Capellini, Yingguang Frank Chan, Conceptualization, Data curation, Formal analysis, Investigation, Writing—review and editing, Acquisition of data; Catherine A Guenther, Data curation, Formal analysis, Investigation, Writing—review and editing, Acquisition of data, Analysis and interpretation of data; Carlos R Infante, Data curation, Formal analysis, Investigation, Analysis and interpretation of data; Douglas B Menke, Conceptualization, Data curation, Formal analysis, Funding acquisition, Investigation, Visualization, Writing—review and editing, Analysis and interpretation of data; David M Kingsley, Conceptualization, Supervision, Funding acquisition, Investigation, Methodology, Writing—original draft, Project administration, Writing—review and editing, Analysis and interpretation of data

### Author ORCIDs

Terence D Capellini (iD) https://orcid.org/0000-0003-3842-8478
Yingguang Frank Chan (iD) https://orcid.org/0000-0001-6292-9681
Douglas B Menke (iD) https://orcid.org/0000-0002-7109-1451
David M Kingsley (iD) http://orcid.org/0000-0002-6401-6461

### Ethics

Animal experimentation: This study was performed in accordance with the recommendations in the Guide for the Care and Use of Laboratory Animals of the National Institutes of Health. All of the animals were handled according to approved institutional animal care and use committee (IACUC)

protocols (#13834, #10665) of Stanford University, in animal facilities accredited by the Association for Assessment and Accreditation of Laboratory Animal Care International (AAALAC).

## Decision letter and Author response

Decision letter https://doi.org/10.7554/eLife.38555.022
Author response https://doi.org/10.7554/eLife.38555.023

## Additional files

### Supplementary files

• Supplementary file 1. Excel file with Supplementary Tables A though G. (**A**) LacZ expression patterns in transgenic embryos carrying BAC sequences. Because only a relatively small number of embryos were obtained in injections with large BAC clones, positive patterns of expression should still be confirmed by additional studies of particular genomic regions, as done in this study for the *PelB* region. (**B**) Putative enhancer regions interacting with the *Pitx1* promoter during mouse development. Andrey et. al used Capture-C methods and histone modification patterns to study chromatin interactions surrounding 446 genes during forelimb and hind limb development at three stages of mouse embryonic development, and in midbrain (*Andrey et al., 2017*). This table summarizes predicted enhancer regions that interact with the *Pitx1* promoter, and how these regions overlap with various genomic sequences that have now been tested for functional activity using transgenic reporter constructs or knockout mice. The nomenclature used for different functionally tested regions in the current and previous studies, and the corresponding mouse genomic coordinates (mm9) and references are listed at the bottom of the table. (**C**) Comparison of *Pitx1* and *Pcdb2* expression in $Pitx1^{+/+}$ and $Pitx1^{PelB-/PelB-}$ E12.5 hind limbs. Summary of four qRT-PCR assays showing the average relative level of *Pitx1* and *Pcdb2* gene expression in $Pitx1^{+/+}$ (n = 7) and $Pitx1^{PelB-/PelB-}$ (n = 8) hind limbs following normalization to the reference gene *Pgk1*. The analysis was carried out with two independent sets of PCR primers for each target gene (*Supplementary file 1G*), and each primer set was used for replicate assays on the fifteen wild type and mutant RNA samples. SD, standard deviation; SEM, standard error of the mean. Only *Pitx1* expression was significantly reduced in *PelB* mutant limbs, *p<0.05. (**D**) $Pitx1^{+/+}$ vs. $Pitx1^{PelB-/PelB-}$ mice. $Pitx1^{+/+}$ n=21, $Pitx1^{PelB-/PelB-}$ n = 20. Yellow = p < 0.01, Orange = p < 0.001, Red = p < 0.0001. (**E**) $Pitx1^{+/+}$ vs. $Pitx1^{null/+}$ mice. $Pitx1^{+/+}$ n=20, $Pitx1^{null/+}$ n=22. Yellow = p < 0.01, Orange = p < 0.001, Red = p < 0.0001. (**F**) $Pitx1^{null/+}$ vs. $Pitx1^{null/PelB-}$ mice. $Pitx1^{null/+}$ n=22, $Pitx1^{null/PelB-}$ n = 22. Yellow = p < 0.01, Orange = p < 0.001, Red = p < 0.0001. (**G**) Primers used in this study.
DOI: https://doi.org/10.7554/eLife.38555.013

• Transparent reporting form
DOI: https://doi.org/10.7554/eLife.38555.014

## Data availability

The data generated and analyzed during this study are included in the manuscript and supporting files, or can be visualized on the UCSC genome browser http://genome.ucsc.edu

The following previously published datasets were used:

| Author(s) | Year | Dataset title | Dataset URL | Database and Identifier |
|---|---|---|---|---|
| Rosenbloom KR, Sloan CA, Malladi VS, Dreszer TR, Learned K, Kirkup VM, Wong MC, Maddren M, Fang R, Heitner SG, Lee BT, Barber GP, Harte RA, Diekhans M, Long JC, Wilder SP, Zweig AS, Karolchik D, Kuhn RM, Haussler D | 2013 | ENCODE data in the UCSC Genome Browser: year 5 update. | https://www.encodeproject.org/experiments/ENCSR000CNB/ | ENCODE, ENCSR000CNB |

| | | | | |
|---|---|---|---|---|
| Andrey G, Schöpflin R, Jerković I, Heinrich V | 2017 | Characterization of hundreds of regulatory landscapes in developing limbs reveals two regimes of chromatin folding | https://www.ncbi.nlm.nih.gov/geo/query/acc.cgi?acc=GSE84795 | NCBI Gene Expression Omnibus, GSE84795 |
| Rosenbloom KR, Sloan CA, Malladi VS, Dreszer TR, Learned K, Kirkup VM, Wong MC, Maddren M, Fang R, Heitner SG, Lee BT, Barber GP, Harte RA, Diekhans M, Long JC, Wilder SP, Zweig AS, Karolchik D, Kuhn RM, Haussler D | 2012 | ENCODE data in the UCSC Genome Browser: year 5 update. | https://www.encodeproject.org/experiments/ENCSR000CNF/ | ENCODE, ENCSR000CNF |

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
