## [Decision Letter]

Thank you for submitting your article "A novel enhancer near the *Pitx1* gene influences development and evolution of pelvic appendages in vertebrates" for consideration by eLife. Your article has been reviewed by three peer reviewers, and the evaluation has been overseen by Marianne Bronner as the Senior and Reviewing Editor. The following individuals involved in review of your submission have agreed to reveal their identity: Neil Shubin (Reviewer #1); Karen E Sears (Reviewer #3). A further reviewer remains anonymous.

The reviewers have discussed the reviews with one another and the Reviewing Editor has drafted this decision to help you prepare a revised submission.

Summary:

The authors have previously shown that mutations in regulatory regions of *Pitx1* are associated with loss of pelvic appendages. This study identifies a novel enhancer, PELB, in mouse that recapitulates some (but not all) elements of *Pitx1* hindlimb expression. Deletion of the enhancer produces some subtle effects on the length of some hindlimb bones. Mutation of the equivalent regulatory region in sticklebacks is associated with reduction in pelvic structures.

While all reviewers found the working interesting, they also raised several issues detailed below that need to be addressed by further experiments and rewriting. I have also included the full review of Reviewer 2 for further details.

Essential revisions:

1) It is clear from this work (and past work) that multiple *Pitx1* enhancers contribute to HL expression in mouse. A concern with the present manuscript is that the approach doesn't allow one to see the diversity of *Pitx1* enhancers that might be involved. This should be clear in the manuscript.

2) The authors should discuss how the skeletal phenotypes match or don't match the enhancer expression-skeletal abnormalities in the anterior compartment but expression is predominantly in the posterior autopod. It is also worth noting the discontinuity between the mild phenotypes observed in the mouse and the significance that is placed on the enhancer for normal development and evolution of pelvic appendages. Some data showing the effect of *PelB* deletion on normal expression levels would be very helpful to clarify.

3) Discussion of the pectoral expression domains and the implications this has for normal control of *Pitx1* expression and evolution of pectoral/pelvic differences.

*Reviewer #2:*

The RP23-55J18 fragment does not produce the broad hindlimb-restricted domain of the endogenous *Pitx1* expression pattern. The *PelB* smaller fragment contained within RP23-55J18 retains the hindlimb expression domain which is further refined into *PelBCon1* and *Con2*. The expression domain is largely restricted to the posterior autopod.

The effect of *PelB* deletion on expression of *Pitx1* is not described (presumably it was too subtle to discern?) but instead the authors document a reduction in length of tarsal and metatarsal bones. The deletion of *PelB* in mouse does not produce a profound reduction in hindlimb formation. The authors have gone to great lengths to produce statistically significant data but reductions ranging from 2-4% reduction in length are subtle nonetheless.

Absence of the enhancer on the background of heterozygous mutation of the *Pitx1* gene can exacerbate the patella reduction seen in the *Pitx1* het. The effect on the patella is puzzling given the enhancer fragment is shown to have activity restricted to the autopod. How does loss of *Pitx1* expression in this domain contribute to patella reduction? Changes in tarsal and metatarsal bones analysed in the *PelB* homozygous mutant have also been analysed in the *pitx1* het, *PelB* het compound mutant. Results are tabulated in Supplementary file 1F only (could they be in an expanded Figure 3?). This background appears to produce reduction in bones lengths similar to that seen in the *PelB* homozygous mutant. Would this be predicted or a more profound effect be expected?

Mutation of the equivalent regulatory region in sticklebacks is associated with reduction in pelvic structures (the authors make the point that the PAXB population show the most extreme reduction in pelvic loss). The effect on mouse hindlimb formation, however, is very subtle and the domain of expression it is capable of producing in transgenics would indicate this element is not as critical as the authors suggest. Is it possible the PAXB population also carry other deleterious mutations to this locus?

I did not find the discussion of the proposal of Osterwalder 2018 helpful as I don't believe the data presented can help clarify a debate on whether multiple functionally redundant enhancers are operating or the converse. This paper describes the isolated case of regulation of a single gene, *PItx1*.

---

## [Author Response]

Essential revisions:1) It is clear from this work (and past work) that multiple Pitx1 enhancers contribute to HL expression in mouse. A concern with the present manuscript is that the approach doesn't allow one to see the diversity of Pitx1 enhancers that might be involved. This should be clear in the manuscript.

We have clarified the description of multiple *Pitx1* enhancers in the Introduction, Results, and Discussion. We also modified Figure 1, and added a new table (Supplementary file 1B), to show the positions of various enhancers described in the literature.

2) The authors should discuss how the skeletal phenotypes match or don't match the enhancer expression-skeletal abnormalities in the anterior compartment but expression is predominantly in the posterior autopod. It is also worth noting the discontinuity between the mild phenotypes observed in the mouse and the significance that is placed on the enhancer for normal development and evolution of pelvic appendages. Some data showing the effect of PelB deletion on normal expression levels would be very helpful to clarify.

We have added a new paragraph discussing different mechanisms by which skeletal phenotypes may arise from the posterior limb expression domain (Discussion, third paragraph). We have also carried out new experiments showing that loss of *PelB* does lead to a reproducible but mild reduction in the levels of *Pitx1* in hindlimbs (subsection “Deletion of *PelB* enhancer in mouse”, first paragraph, Discussion, second paragraph, subsection “qRT-PCR”, new table (Supplemental file 1C), and see additional discussion below).

3) Discussion of the pectoral expression domains and the implications this has for normal control of Pitx1 expression and evolution of pectoral/pelvic differences.

We have enlarged our discussion of the *PelB* mutation found in the extremely pelvic-reduced PAXB stickleback population, including the fact that there are no obvious morphological differences seen in the pectoral or median fins of PAXB fish compared to other populations that also show pelvic reduction (Discussion, tenth paragraph). We have also added a new paragraph that compares the time-scale of enhancer evolution in the *Pitx1* gene with the larger evolution of pectoral/pelvic differences in vertebrates (Discussion, eighth paragraph, and see further discussion below).

Reviewer #2:

The RP23-55J18 fragment does not produce the broad hindlimb-restricted domain of the endogenous Pitx1 expression pattern. The PelB smaller fragment contained within RP23-55J18 retains the hindlimb expression domain which is further refined into PelBCon1 and Con2. The expression domain is largely restricted to the posterior autopod.The effect of PelB deletion on expression of Pitx1 is not described (presumably it was too subtle to discern?) but instead the authors document a reduction in length of tarsal and metatarsal bones. The deletion of PelB in mouse does not produce a profound reduction in hindlimb formation. The authors have gone to great lengths to produce statistically significant data but reductions ranging from 2-4% reduction in length are subtle nonetheless.

As noted above, we now describe changes in both *Pitx1* gene expression and morphology in the *PelB* mutant mice.

Absence of the enhancer on the background of heterozygous mutation of the Pitx1 gene can exacerbate the patella reduction seen in the Pitx1 het. The effect on the patella is puzzling given the enhancer fragment is shown to have activity restricted to the autopod. How does loss of Pitx1 expression in this domain contribute to patella reduction? Changes in tarsal and metatarsal bones analysed in the PelB homozygous mutant have also been analysed in the pitx1 het, PelB het compound mutant. Results are tabulated in Supplementary file 1F only (could they be in an expanded Figure 3?). This background appears to produce reduction in bones lengths similar to that seen in the PelB homozygous mutant. Would this be predicted or a more profound effect be expected?

As noted above, in the revised manuscript, we have added a new paragraph describing different mechanisms by which loss of expression in the posterior domain could give rise to patella and more anterior phenotypes (Discussion, third paragraph). We also list several approaches that could be used in the future to further study this question (including lineage tracing of *PelB* expressing cells, testing whether there are larger changes in *Pitx1* chromatin structure in the absence of *PelB*, and testing whether *PelB* mutations also alter other tissues like muscle).

Mutation of the equivalent regulatory region in sticklebacks is associated with reduction in pelvic structures (the authors make the point that the PAXB population show the most extreme reduction in pelvic loss). The effect on mouse hindlimb formation, however, is very subtle and the domain of expression it is capable of producing in transgenics would indicate this element is not as critical as the authors suggest. Is it possible the PAXB population also carry other deleterious mutations to this locus?

PAXB fish have completely lost another pelvic enhancer of the *Pitx1* locus (*PelA*, Chan et al., 2010), in addition to the novel mutation that we now describe for the first time in *PelB*. The revised Discussion further emphasizes that it may be the loss of both *PelA* and *PelB* enhancers that produces the more severe in PAXB fish, compared to other known fish that have only lost *PelA* (Discussion, tenth paragraph). We also outline how this could be further studied in the future by making engineered mutations in *PelA, PelB*, or both enhancers together in fish (Discussion, last paragraph).

I did not find the discussion of the proposal of Osterwalder 2018 helpful as I don't believe the data presented can help clarify a debate on whether multiple functionally redundant enhancers are operating or the converse. This paper describes the isolated case of regulation of a single gene, PItx1.

Although our paper describes a detailed study of the single locus *Pitx1*, we believe the results are relevant to the broader Osterwalder proposal of "redundant enhancers" existing in many genes. In our Discussion, we are also careful to draw out general parallels between our own findings with the *Pitx1* locus and comparable expression and genetic knockout studies of multiple hindlimb enhancers that also exist in the *Tbx*4 gene (Discussion, fifth and sixth paragraphs). The points made thus have wider significance than just a single locus, and we think the results add a useful additional perspective to the literature.